https://doi.org/10.1038/s41467-018-07972-7　**OPEN**

# How lasing happens in CsPbBr$_3$ perovskite nanowires

Andrew P. Schlaus[1], Michael S. Spencer[1], Kiyoshi Miyata [1], Fang Liu[1], Xiaoxia Wang[2], Ipshita Datta [3], Michal Lipson[3], Anlian Pan[2] & X.-Y. Zhu[1]

Lead halide perovskites are emerging as an excellent material platform for optoelectronic processes. There have been extensive discussions on lasing, polariton formation, and non-linear processes in this material system, but the underlying mechanism remains unknown. Here we probe lasing from CsPbBr$_3$ perovskite nanowires with picosecond (ps) time resolution and show that lasing originates from stimulated emission of an electron-hole plasma. We observe an anomalous blue-shifting of the lasing gain profile with time up to 25 ps, and assign this as a signature for lasing involving plasmon emission. The time domain view provides an ultra-sensitive probe of many-body physics which was obscured in previous time-integrated measurements of lasing from lead halide perovskite nanowires.

[1] Department of Chemistry, Columbia University, New York, NY 10027, USA. [2] Key Laboratory for Micro-Nano Physics and Technology of Hunan Province, College of Materials Science and Engineering, Hunan University, Changsha 410082, China. [3] Department of Electrical Engineering, Columbia University, New York, NY 10027, USA. Correspondence and requests for materials should be addressed to X.-Y.Z. (email: xyzhu@columbia.edu)

Lead halide perovskites (LHPs) continue to draw attention for their extraordinary photovoltaic efficiencies and their expanding roles in optoelectronic research. Light emission with near unity quantum yield, low lasing thresholds, and compositionally tuneable wavelength makes them strong contenders for highly efficient light emitting devices, nanowire (NW) lasers, and potentially exciton-polariton devices[1–5]. Photophysical studies in the past few years have established that charge carrier properties in LHPs are distinct from those in conventional semiconductors; those in the former are exemplified by exceptional defect tolerance, slow hot carrier cooling, and efficient dynamic screening[6]. Despite a plethora of publications on carrier dynamics[6–14] and lasing[1,15–20] in LHPs, it remains unclear how these two aspects are related.

Central to the debate on the lasing mechanisms is the role of excitons. Various mechanisms have been proposed to explain the quantitative characteristics of lasing from NW or other microcavities of LHPs[1,15–20]. The formation of exciton-polaritons, a coherent superposition between an exciton and a photon in a microcavity, is well known in layered LHPs[21–23] and has been suggested as an underlying lasing mechanism in LHPs[16,20]. While exciton-polaritons may exist at low excitation density and continuous wave (CW) conditions, lasing under pulsed excitation may occur above the exciton Mott density from stimulated emission from a non-degenerate electron hole plasma (n-EHP, also referred to as a Coulomb-correlated EHP)[24–27]. Here, we use ultrafast time-resolved photoluminescence (PL) to directly probe lasing dynamics in $CsPbBr_3$ perovskite NWs via spectral evolution with ~1 ps time resolution. We carry out complementary measurements through ultrafast transient reflectance. We find that the time-integrated laser emission spectra, typical in nearly all reports on LHP lasers published to date, obscure the intrinsic nonlinear physics in the system. Rather, the lasing spectrum under pulsed excitation is a strongly time-dependent function, which consists of red-shifting cavity modes concurrent with blue-shifting laser gain profiles. The latter is unprecedented, and is strong evidence for stimulated emission from an n-EHP coupled with plasmon emission.

## Results

**Nanowire samples**. We use single crystal $CsPbBr_3$ NWs grown from vapor deposition on sapphire substrates[28]. These NWs are of triangular cross-section with hundreds of nanometers lateral dimensions and with lengths in the tens of microns range, as shown by scanning electron microscope (SEM) images in Fig. 1a. We find that these vapor-grown NWs are more stable under optical excitation than solution-grown NWs used in our previous studies[16,17], permitting experiments at higher excitation densities in the current report. We carry out electromagnetic (EM) wave analysis using the COMSOL Multiphysics finite element method (FEM), as detailed in Supplementary Note 4. A representative simulation result for the lowest energy waveguided mode is shown in Fig. 1b, and the full set of simulated mode profiles and effective refractive indices are shown in Supplementary Figure 8 and Supplementary Figure 9. In the experiment, Fig. 1c, an individual $CsPbBr_3$ NW at a sample temperature of 80 K is uniformly excited by a short laser pulse (~60 fs) with photon energy ($\hbar\omega = 3.1$ eV) above the bandgap ($E_g = 2.35$ eV). We time-resolve the fluorescence and laser emission spectra from the photo-excited NW using a Kerr-gating technique, as illustrated in Fig. 1c (see Methods). Figure 1d shows the lasing mechanism deduced from the time-resolved measurements, as detailed below. In the following (Figs. 2 and 3), we show results from a single NW of 15 μm length. Additional representative lasing results and discussion from other NWs and nanoplate samples can be found

in Supplementary Figure 1, Supplementary Figure 2, and Supplementary Note 1.

**Excitation density-dependent lasing**. We begin our investigation by analyzing the PL spectrum as a function of incident excitation laser fluence ($\rho$, pulse energy per unit area), as shown in a two-dimensional (2D) pseudo-color plot in Fig. 2a and as horizontal cuts of the pseudo-color plot in Fig. 2b. Here, the power density can be converted to excitation density based on 1 μJ cm$^{-2}$ = $1.6 \times 10^{23}$ m$^{-3}$, as determined by the reflectivity of the sample and a unity absorption coefficient and sample illumination geometry. At $\rho < 2$ μJ cm$^{-2}$, PL shows fluence-independent spectral shape with a maximum at $2.357 \pm 0.001$ eV, in correspondence with that of spontaneous emission from crystalline $CsPbBr_3$ perovskite[29]. The integrated PL intensity scales with $\rho^\alpha$, with $\alpha = 1.5 \pm 0.1$, in this low fluence region (see fit, black line, to the red circles in Fig. 2c). PL emission from the radiative recombination of electrons and holes is a second order ($\alpha = 2$) process, as is observed for single crystal $CsPbBr_3$ at room temperature[30]. In contrast, PL emission from excitons is a first-order process ($\alpha = 1$). The $\alpha = 1.5 \pm 0.1$ value determined here for PL emission from $CsPbBr_3$ at 80 K may be attributed to radiative recombination from electron and hole carriers in the presence of a less radiative population, e.g., partial indirect bandgap character, an equilibrium between free carriers and excitons, large polaron formation, or competitive trapping[31].

At $\rho \geq 3$ μJ cm$^{-2}$, we observe the emergence of a group of regular and narrow peaks on the lower energy side of the PL peak maximum (Fig. 2a, b) and an increase in the rate of growth of PL intensity with respect to $\rho$ in this spectral region, blue circles in Fig. 2c. The appearance of these peaks corresponds to lasing, as reported earlier[15–17,19]. With increasing $\rho$, more lasing peaks appear on the lower energy side of the PL spectra. Unlike previous suggestions of exciton-polariton origins[16,20], we find that lasing comes from a Coulomb correlated EHP, i.e., n-EPP[7], as the calculated excitation density at the lasing threshold of 3 μJ cm$^{-2}$ is $4.8 \times 10^{23}$/m$^3$, which is above the exciton Mott-density (see Fig. 5c below and Supplementary Note 5). The appearance of lasing peaks only on the lower energy side of the PL peak, not at the PL peak where oscillator strength is the highest, is consistent with stimulated emission from an n-EHP with the simultaneous emission of a plasmon[24,26,32]. We observe a saturation in lasing intensity at $\rho_2 = 30$ μJ cm$^{-2}$ or excitation density of $4.8 \times 10^{24}$/m$^3$, above which the spectral density shifts back toward the band edge. The increased screening above $\rho_2$ diminishes the Coulomb correlations as the system transitions into a degenerate EHP[26]; as a result, the transition cross-section for lasing is decreased and this is responsible for the saturation behavior. Further insight into the EHP lasing mechanism comes from the evolution of the lasing spectral profile with time (Fig. 3) and with excitation density (Fig. 2), as detailed below.

**Ultrafast photoluminescence of nanowire lasers**. We probe the laser emission with picosecond time resolution to directly monitor the lasing dynamics, Fig. 3a–c. We make four observations: (1) the onset of lasing occurs with time delay $\Delta t_1 = $ ~1–3 ps, (2) the initially broad lasing spectrum narrows in $\Delta t_2 = $ ~3–7 ps, (3) on longer time scales of $\Delta t_3 = 5$–30 ps, the lasing profile climbs to higher energies and moves closer to the PL peak with increasing time, and (4) throughout the experimental time window, each lasing mode red-shifts with time. For observations (1)–(3), both the onset time and the duration for each step increase with increasing excitation density. The time-resolved results in Fig. 3 show that the broad lasing peaks in Fig. 2 do not reflect the intrinsic linewidths of the lasing peaks, but instead arise from the

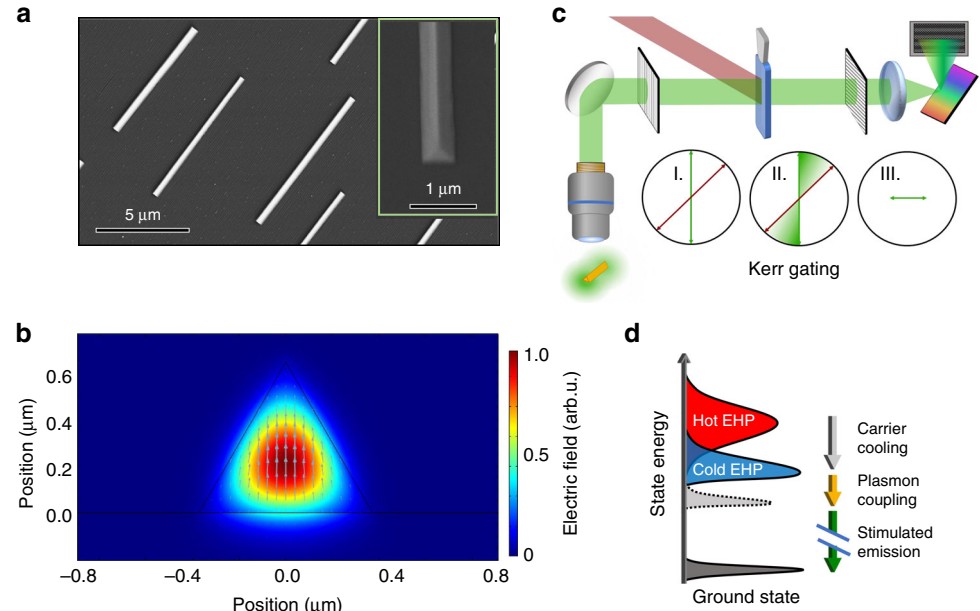

**Fig. 1** Nanowire samples, experimental setup, and lasing mechanism. **a** Scanning electron microscopy images of triangular nanowires grown on sapphire substrate. **b** FEM simulation of the lowest order waveguiding mode in a nanowire. The electric field polarization is depicted by the cyan arrows. **c** Illustration of the optical setup for time-resolved Kerr gating experiment. A microscope is used for excitation of the nanowire and collection of the lasing emission. The linear polarization (I) becomes elliptical (II) as it passes through the Kerr medium with the pump pulse. A final polarizer (III) filters polarization perpendicular to the original incident polarization. **d** Cartoon describing the carrier dynamics from photoexcitation which results in a hot electron hole plasma (Hot EHP) through carrier cooling to a cold electron hole plasma (Cold EHP) finishing with stimulated emission coupled with plasmon emission

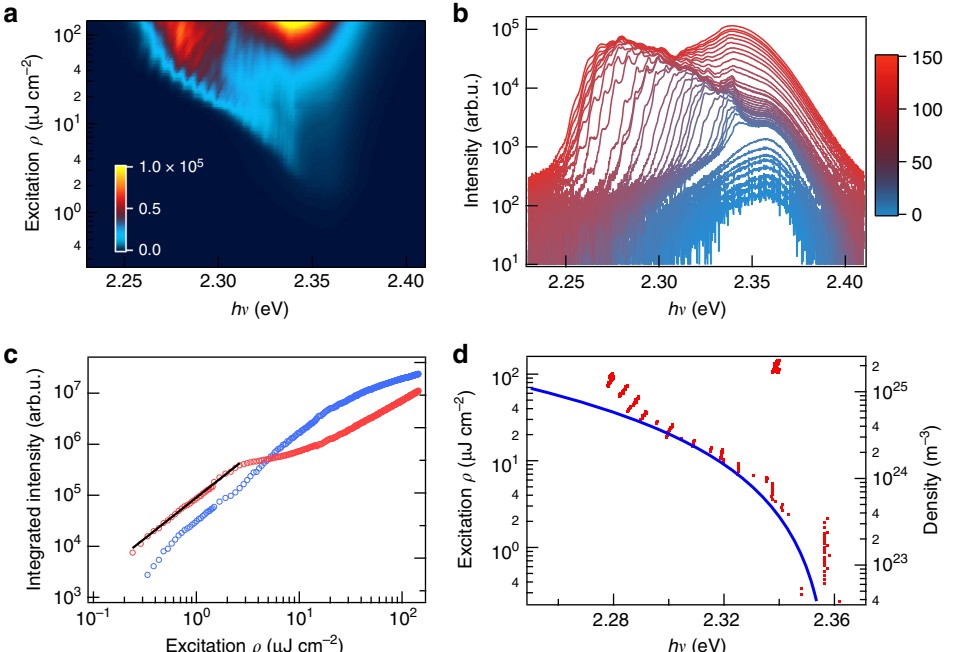

**Fig. 2** Excitation density-dependent lasing spectra revealing lasing and saturation thresholds. Two-dimensional pseudo-color plot (**a**) and horizontal cuts (**b**) of photoluminescence spectra as a function of increasing excitation energy density ($\rho$), 0.24–145 μJ cm$^{-2}$. The excitation energy density in (**b**) increases from blue to red (0.43, 0.53, 0.62, 0.77, 1.0, 1.1, 1.3, 1.5, 2.4, 2.6, 2.9, 3.1, 3.6, 4.3, 5.3, 6.5, 7.2, 7.9, 8.6, 9.6, 11.0, 12.5, 13.9, 15.4, 17.3, 19.7, 21.6, 24.0, 33.1, 37.9, 49.9, 61.9, 71.5, 83.5, 95.5, 107.5, 119.5, 131.5, 144.0 μJ cm$^{-2}$). Note the logarithmic scale for emission intensity in (**b**). These spectra show the evolution of emission from below the lasing threshold (3 μJ cm$^{-2}$) through the lasing saturation threshold (30 μJ cm$^{-2}$) and above. Stimulated behavior is confirmed from the PL intensity as a function of excitation density (**c**), showing the integrated intensity in the lasing spectral region (blue) where the onset of lasing corresponds to superlinear behavior and saturation of the PL intensity (red). An exponential fit (black line) represents the power scaling law of $\rho^{1.5}$ present below the lasing threshold. As pump fluence increases, the lasing spectral density red-shifts, as shown by the positions (red dots) of the most intense peak in lasing/PL spectra as a function of excitation density (**d**). The blue curve in (**d**) shows fit to the excitation density-dependent plasmon energy. All spectra were obtained from a single NW with 15 μm length at a substrate temperature of 80 K

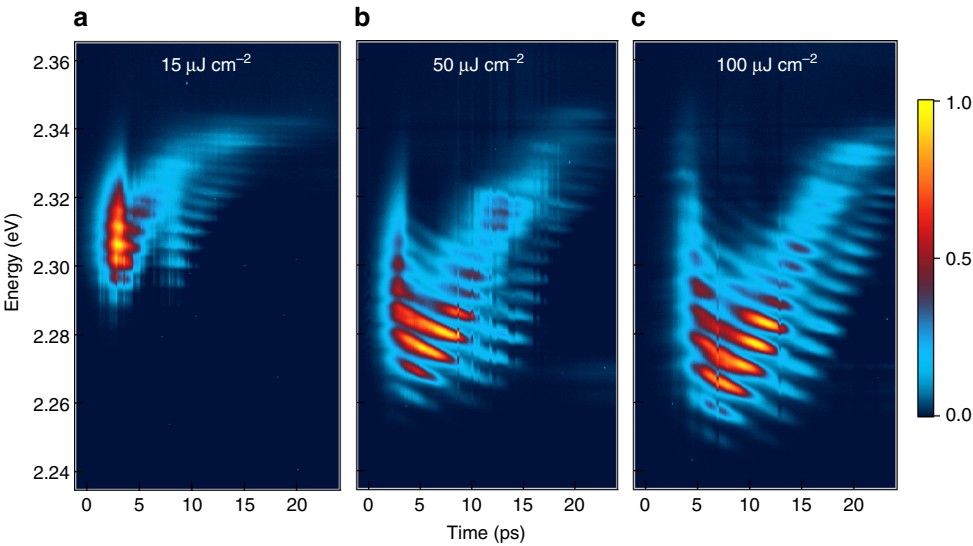

**Fig. 3** Time-resolved lasing. The 2D pseudo-color (normalized intensity) plots show emission spectra at **a** 15, **b** 50, and **c** 100 μJ cm$^{-2}$. These powers represent the lasing region before saturation, the saturation region, and the high power limit. Line-cuts and integrated spectra can be found in Supplementary Figure 4. All spectra were obtained from a single NW with 15 μm length at a substrate temperature of 80 K

time integration of spectrally narrower peaks shifting with time or excitation density. These effects are also obvious in horizontal cuts at selected delay times and in the comparison of time-resolved and time-integrated spectra, Supplementary Figure 3. For further discussion, see Supplementary Note 2. These results are all consistent with stimulated emission from an n-EHP.

An EHP is inherently a two-level electronic system for which population inversion is difficult. As first proposed by Klingshirn and coworkers for lasing in ZnO NWs[24,26,32], stimulated photon emission from an n-EHP is coupled to the emission of a plasmon quanta, i.e., collective oscillations of an EHP, shown schematically in Fig. 1d. This coupling introduces intermediate states and creates a situation reminiscent of a classic three-level lasing scheme[33]. Critically, the addition of coupling to a plasmon resonance relaxes the criterion for population inversion in the semiconductor, traditionally given by[34]:

$$\mu_e + \mu_h > E_{e,c}(k) + E_{h,v}(k), \tag{1}$$

where $\mu_e$, $\mu_h$ stand for electron and hole chemical potentials, and $E_{e,c}$, $E_{h,v}$ stand for electron and hole kinetic energies in the valence and conduction bands, respectively; they are both understood to be zero at the band edge. In the presence of plasmon-coupled emission, the inversion criterion is instead given by:

$$\mu_e + \mu_h > E_{e,c}(k) + E_{h,v}(k) - \hbar\omega_p, \tag{2}$$

where $\omega_p$ is the plasmon frequency. This relaxation of lasing requirements gives rise to both the sub-bandgap lasing spectra and lasing threshold below the conventional inversion threshold determined by Eq. (1).

The plasmon frequency of an electron plasma may be approximated by:

$$\omega_p = \sqrt{\frac{n_e e^2}{m^* \varepsilon_{\text{eff}} \varepsilon_o}}, \tag{3}$$

where $n_e$ and $m^*$ are the number density and band mass of electrons in the conduction band, $\varepsilon_{\text{eff}}$ is the effective dielectric constant which is frequency and carrier density-dependent, $\varepsilon_o$ is vacuum permittivity. The same equation applies to the hole ($n_h$) in the valence band. The red-shift can thus be understood as a

measure of the plasmon energy, $\hbar\omega_p$, which scales with $\sqrt{n_e}$. The total red-shift in lasing peak positions (red dots in Fig. 2d) is $\Delta_p = -80$ meV, as the pump fluence increases from the lasing threshold 3 μJ cm$^{-2}$ to the saturation threshold ~30 μJ cm$^{-2}$. The estimated density-dependent $\hbar\omega_p$ (blue curve in Fig. 2d) is in agreement with experimental data. To approximately describe the data, we use a constant effective dielectric constant $\varepsilon_{\text{eff}} = 19.2$, which is 4× the optical dielectric constant of CsPbBr$_3$[6]. The large increase in $\varepsilon_{\text{eff}}$ is expected from the Drude-like response of a highly excited semiconductor in the EHP region.

The time-dependent lasing profiles in Fig. 3 are in excellent agreement with the n-EHP model, and provide a direct view of the many-body dynamics. Following initial photoexcitation, formation of hot electron/hole distributions occur on the ultra-short time scales of tens of femtoseconds[35] and this is not resolved within our time resolution (~1 ps). The establishment of e–h correlations, are commensurate with hot carrier cooling via LO-phonon-lattice thermalization, which occurs on the slightly longer time scale of $t_{\text{corr}} \sim 1$ ps[36]; this process increases the oscillator strength and is responsible for the onset time ($\Delta t_1$) in the appearance of lasing spectrum. The observed increase of $\Delta t_1$ with excitation density is consistent with density-dependent $t_{\text{corr}}$ and the slower cooling at higher carrier density[10–14].

Each lasing spectral profile is a product of the gain profile and the photonic modes defined by the natural Fabry–Perot cavity. Following excitation, we observe a short-lived and broad lasing spectrum, and thus the gain profile can be attributed to hot carriers before they are completely thermalized with the phonon bath. The hot lasing profile narrows down to several predominant modes at the low energy end. This is expected from the cooling of hot carriers and increasing magnitude of electronic correlations in the plasma. The lengthening in time duration ($\Delta t_2$) of this process with increasing excitation density is also in agreement with the slower cooling at higher densities[10–14].

As the cooling process continues, the laser gain profile blue-shifts and moves closer to the PL peak energy with time. To our knowledge, this observation is unprecedented in NW lasing literature. In the much studied ZnO NW lasing literature, red-shifts in lasing modes over time has been reported and analyzed[27]. In published spectra from one time-resolved study on ZnO NWs, there was evidence of blue-shift on the tens of

picoseconds time scale but the authors did not provide an analysis or interpretation[37]. The blue-shifting effect is opposite to the initial red-shift attributed to hot carrier cooling. The blue-shift on the tens of picoseconds time scale in our time-resolved measurement can in fact be taken as signature for plasmon-coupled lasing from an n-EHP, dictated by Eq. (3). As the carrier density is depleted with increasing time via stimulated emission, the plasmon energy ($\hbar\omega_p$) decreases. At an initial excitation density above the laser saturation threshold (~30 μJ cm$^{-2}$), the plasmon energy blue-shifts by $\Delta\hbar\omega_p \sim 60$ meV over the 25 ps time window probed here. The extent of total blue-shift increases with initial excitation density from the lasing to the saturation threshold, as is shown by a comparison of Fig. 3a, b. In the saturation regime, Fig. 3b, c, we no longer observe significant spectral changes, rather the increased screening at higher excitation density prolongs each step of the time-dependent laser spectral evolution.

Present throughout all excitation densities above the lasing threshold is the temporal red-shift of individual lasing modes. The cavity geometry is fixed by the NW but the refractive index, $n_r$, around the bandgap decreases with increasing excitation density[34,38]. Since the energy of each cavity mode ($j$) is inversely proportional to refractive index, $E_j \propto 1/n_r$, the mode energy decreases as $n_r$ recovers with time, and the carrier density is depleted. Moreover, a higher initial excitation density corresponds to a larger initial decrease in $n_r$; this gives a steeper slope,

$-dE_j/dt$, in the red-shift of mode energy, as is observed experimentally in Fig. 3. Similarly, the carrier density-dependent $n_r$ also accounts for the blue-shift in each lasing mode with $\rho$ in Fig. 2. Supplementary Figures 4–6 measure this quantitative spectra-shifting of the lasing modes and is further described in Supplementary Note 3.

**Transient reflectance of bulk single crystal CsPbBr$_3$.** To confirm the nature of the excitations and electronic phase transitions, we carry out transient reflectance ($\Delta R/R$) measurement on CsPbBr$_3$ single crystals, Fig. 4. We use conditions close to those in NW lasing experiments. The sample was cooled to 80 K, excited across the bandgap at $\hbar\omega = 3.1$ eV, and probed by a white light pulse. Below the lasing threshold at $\rho = 0.6$ μJ cm$^{-2}$ in 2D plot in Fig. 4a or horizontal cuts at $\Delta t = 50$ fs for $\rho = 0.015$–1.2 μJ cm$^{-2}$ in Fig. 4e, we observe an antisymmetric peak, which has been observed and analyzed before in transient reflectance studies on single crystals CsPbBr$_3$ and its hybrid counterparts[30,39]. The anti-symmetric peak shape results mainly from the frequency dependent and photo-induced change in the refractive index $\Delta n(h\nu)$, which can be obtained at low fluences from the Kramers–Kronig transformation of photo-induced change in the absorption coefficient, $\Delta\alpha(\hbar\omega)$[30,39]. In the low pump fluence region (at $\rho < 3$ μJ cm$^{-2}$), the anti-symmetric peak crosses zero at $\hbar\omega = 2.368 \pm 0.001$ eV, which corresponds to the peak

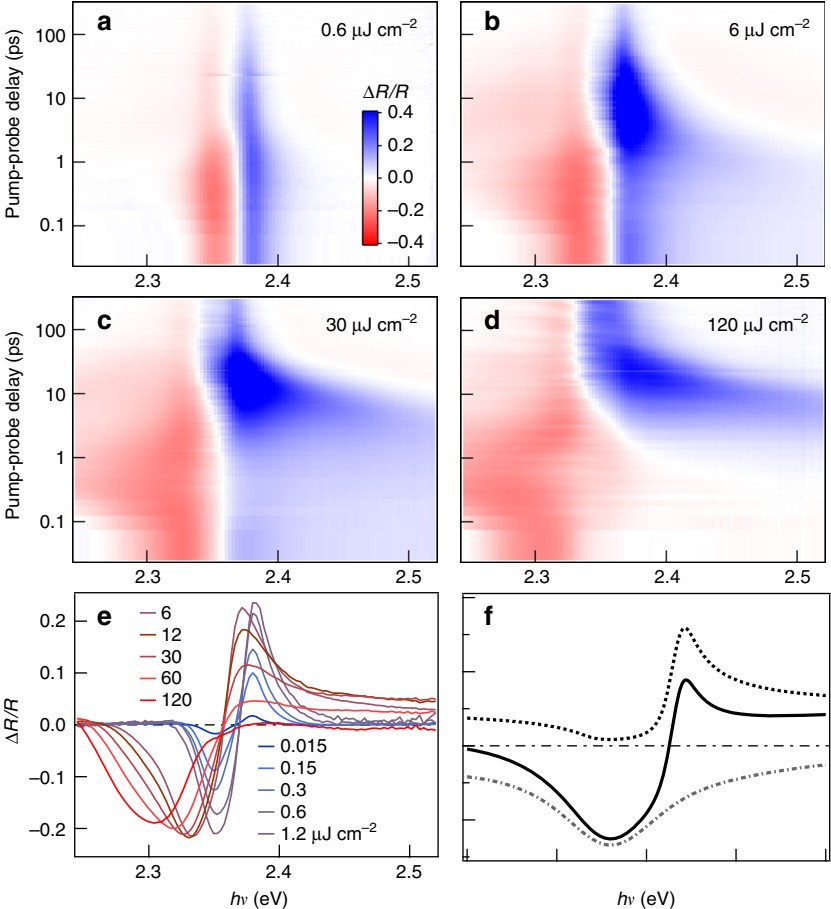

**Fig. 4** Transient reflectance spectra reveal transitions from excitonic resonance to n-EHP and d-EHP. **a–d** 2D pseudo-color plots of transient reflectance spectra from a CsPbBr$_3$ single crystal at excitation densities of 0.6, 6, 30, and 120 μJ cm$^{-2}$, respectively. The excitation photon energy was 3.1 eV and the sample temperature was 80 K. **e** Transients taken at time delay = 50 fs for a variety of pump fluences spanning the regimes of interest. **f** Simulated spectral shapes typical of a plasma (dotted curve) and optical gain (dot-dashed curve) and the sum of the two (solid curve); see Supplementary Note 6 for details on spectral simulation

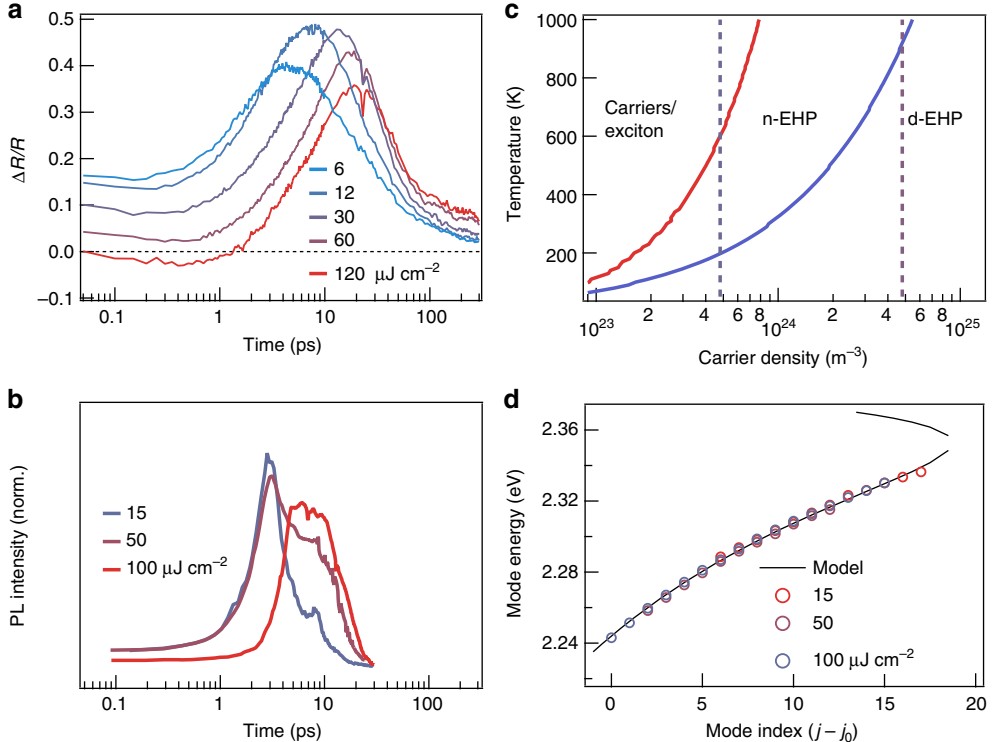

**Fig. 5** Carrier cooling, phase transitions, and mode energies. **a** Time evolution of the positive feature in the transient reflectance representing the carrier cooling dynamics as various pumping powers. **b** Integrated lasing intensity from the plots in Fig. 3, demonstrating the connection with the time scales for carrier cooling in panel (**a**). **c** Phase diagram showing the temperature and carrier densities at which the Mott densities and population inversions lie, leading to the three electronic phase regimes: thermodynamic population of carriers and excitons, nondegenerate electron hole plasma (n-EHP), degenerate electron hole plasma (d-EHP). **d** Experimentally determined modes (extrapolated to the lasing threshold) shown along the mode profile calculated using a single Lorentzian oscillator for the dielectric function (see Supplementary Note 7)

position of the excitonic resonance in the absorption spectrum[39]. This value is very close to the PL peak position ($\hbar\omega = 2.357 \pm 0.001$ eV), indicating a very small Stokes shift. Note that, the anti-symmetric peak shape remains constant for low fluence at $t > \sim3$ ps, it slightly blue-shifts and broadens at shorter times due to initial hot carrier cooling and localization. The full set of transient reflectance spectra can be found in Supplementary Figure 7.

When the excitation density is above the lasing threshold, Fig. 4b, c or horizontal cuts at $\Delta t = 50$ fs for $\rho = 6-120\,\mu J\,cm^{-2}$ in Fig. 4e, the zero crossing points are red-shifted by ~0.01 eV to $2.358 \pm 0.001$ eV. The bandgap renormalization may change in fundamental ways as the system loses excitonic resonance character and enters the n-EHP region. In this high-density region, the reflectance is no longer anti-symmetric. In particular, a positive step-like feature develops above the resonance, in agreement with theoretical simulation. In the simulation, we approximate the band edge as a single oscillator and into account of both reflection and stimulated emission from an EHP, Fig. 4f (see Supplementary Note 6 on simulation details). Below the zero crossing, a negative feature broadens to lower energy at higher excitation densities; this is similar to excitation power dependent lasing seen in the NWs (see Figs. 2 and 3) and is here attributed predominantly to amplified spontaneous emission (ASE), coupled with plasmon emission, from the n-EHP.

While the antisymmetric spectral shape in transient reflectance corresponds to bleaching of the excitonic correlations[39], this feature diminishes for $\rho > 30\,\mu J\,cm^{-2}$ and disappears completely on short time scales at the highest excitation density of $\rho = 120\,\mu J\,cm^{-2}$, consistent with the transition from n-EHP to d-EHP. At each excitation density, the antisymmetric feature, and thus the

electron–hole correlations in the EHP, rise with increasing time. We follow the dynamics of this process from the time dependence of the spectral intensity at ~2.37 eV, which is the positive peak position of the antisymmetric feature. The rise of this signal (blue in Fig. 4b–d) slows down by an order of magnitude as initial excitation density increases from 6 to 120 $\mu J\,cm^{-2}$. Figure 5a shows vertical cuts of the 2D pseudo-color plots at 2.37 eV, including only excitation densities above the lasing threshold (6–120 $\mu J\,cm^{-2}$). We find that the initial reflectivity decreases with increasing excitation density, as expected due to the diminished electron–hole correlations in the EHP. The growth of this signal on the picosecond time scale slows down by one order of magnitude, as the excitation density increases from 6 to 120 $\mu J\,cm^{-2}$. This results from the slowed cooling of hot carriers with increasing excitation density, an effect attributed to phonon bottlenecks and/or low thermal conductivity in LHP[10–14]. The cooling of hot carriers towards the band edges is known to increase the electron–hole correlations, and thus oscillator strength[34]. The impact of carrier cooling rate is also seen in the integrated lasing intensity, Fig. 5b. The onset of lasing occurs on the same time scales as the buildup of carriers near the band edges. Further supporting the above interpretation, we model the thermodynamic phase transitions for the Mott-threshold ($\rho_1$) and the d-EHP threshold ($\rho_2$) as a function of temperature and excitation density, Fig. 5c (see Supplementary Note 5). The experimentally determined lasing threshold ($\rho_1$) and the saturation threshold ($\rho_2$) are shown as dashed lines; they cross the calculated phase boundaries at electronic temperatures of 600 and 900 K, respectively. These electronic temperatures are the upper bounds of the estimated electronic temperatures on the tens of picoseconds time scale from previous spectroscopic measurements at the corresponding excitation densities[12]. Thus, the lasing threshold is

at or above the Mott density for transition to the n-EHP and the saturation thresholds correspond to transition to the d-EHP phase.

## Discussion

A NW laser is distinct from a conventional laser in that the whole NW lasing cavity is the gain medium. This cavity character is responsible for the time-dependent red-shift of each lasing mode (Fig. 3) attributed to the excitation density-dependent refractive index, $n(\rho)$[34]. Another manifestation is the nonlinear mode dispersion reflected in the decreasing energy spacing of the lasing modes as the energy of the modes moves closer to the excitonic resonance, i.e., PL peak. Such a nonlinear dispersion indicates strong light–matter interaction, which has been interpreted previously as due to polaritons in the bottleneck region[16]. In view of the current findings on plasmon-coupled n-EHP lasing mechanism in CsPbBr$_3$ NWs, we now re-interpret this strong light–matter interaction from the frequency dependent refractive index, $n(\omega)$. Considering only the real part of the refractive index, the jth mode in a Fabry–Perot cavity can be approximated by

$$E_j = \frac{hc}{2L}\frac{j}{n(\omega)}. \tag{4}$$

When $\omega$ approaches a resonance from the lower energy side, $n(\omega)$ increases and $\Delta E_j = E_{j+1} - E_j$ decreases. This is obvious in a simple model involving a Lorentzian oscillator embedded in a dielectric medium for the n-EHP; the dielectric function is[40]:

$$\epsilon(\omega) = 1 + \frac{\omega_p^2}{\omega_o^2 - \omega^2 - i\Gamma\omega}, \tag{5}$$

and $n(\omega)$ is related to the dielectric function by:

$$\sqrt{\epsilon(\omega)} = n(\omega) + ik(\omega), \tag{6}$$

where $\omega_0$ is the oscillator frequency, $\Gamma$ is the dephasing rate, and $\omega_P$ is the plasmon frequency; $\hbar\omega_p$ is ~20 meV at the lasing threshold (Fig. 2d). Figure 5d shows the dispersion of lasing modes (open circles) extrapolated to the lasing threshold for three starting excitation densities (see Supplementary Figures 4–6 in Supplementary Information). The nonlinear dispersion with negative mode spacing curvature can be well-described by Eq. (4), black curve in Fig. 5d, numerically solved for initial conditions, with $n(\omega)$ given by a Lorentzian oscillator (see Supplementary Note 7 and Supplementary Figure 10). To summarize, we provide a time-domain view of lasing and many-body interaction in single crystal CsPbBr$_3$ perovskite NWs. These measurements establish that above threshold, lasing in CsPbBr$_3$ NWs is not due to excitons or exciton-polaritons, but to stimulated emission from a nondegenerate electron-hole plasma coupled with plasmons. We show that the lasing mode distribution in the NW is a strong function of excitation density and time (in pulsed operation) due to changes in both laser gain profile and refractive index. While these findings reveal fundamental limitations of a NW as a stable lasing platform, there can be engineering approaches to overcome these obstacles. Examples include employing long excitation pulses and feedback in pumping power to stabilize the shifting lasing modes and coupling the NW to an external optical cavity or photonic structure to select a narrow lasing wavelength window while suppressing all other wavelengths. From a fundamental perspective, the shifting lasing modes in a NW cavity serves as an ultra-sensitive probe of many-body dynamics. While the present study focuses on CsPbBr$_3$ NWs, the conclusions likely apply to other lead halide perovskite systems. Compared to their hybrid organic–inorganic counterparts, CsPbBr$_3$ possesses higher exciton binding energy, and thus smaller exciton radius[41,42]. This indicates that the Mott thresholds in hybrid lead bromide perovskites are lower than that in CsPbBr$_3$ and lasing in these hybrid systems is also expected to be due to stimulated emission from a nondegenerate electron–hole plasma coupled with plasmons.

## Methods

**Time-resolved photoluminescence**. All PL measurements, both static and time-resolved, were performed at a sample temperature of 80 K on sapphire substrates mounted to the copper cold head with silver paste in a cryostat (Cryo Industries of America, RC102-CFM Microscopy Cryostat with LakeShore 325 Temperature Controller). The cryostat was operated at pressures <$10^{-7}$ mbar (pumped by a Varian turbo pump) and cooled with flow through liquid nitrogen. The second harmonic of a Clark-MXR Impulse laser (repetition rate of 0.5 MHz, 250 fs pulses, 1040 nm) was used to pump a home-built non-collinear optical parametric amplifier to generate 800 nm pulses (~60 fs) which was used to generate 400 nm pulses via second harmonic generation. The beam size is expanded to ensure illumination across the entire NW and focused onto the sample using a far-field epifluorescence microscope (Olympus, IX73 inverted microscope) equipped with a ×40 objective with NA 0.6, with correction collar (Olympus LUCPLFLN40X) and a 490 nm long-pass dichroic mirror (Thorlabs, DMPL490R). The emission spectra for both static and time-resolved measurements were collected with a liquid nitrogen cooled CCD (Princeton Instruments, PyLoN 400B) coupled to a spectrograph (Princeton Instruments, Acton SP 2300i) with 1200 mm$^{-1}$ grating blazed at 300 nm. We used the Lightfield software suite (Princeton Instruments) and LabVIEW (National Instruments) in data collection. Data analysis was done in Igor Pro (WaveMetrics).

To time resolve the emission from the NWs, we used a Kerr gating technique. The emission from the NW was passed through a linear polarizer then focused into a cuvette of liquid CS$_2$ noncollinear with a gating pulse supplied by the fundamental of the Clark-MXR Impulse (0.5 MHz, 250 fs pulse width, 1040 nm). As a result of the optical Kerr effect induced within liquid CS$_2$ by the gating pulse, an elliptical polarization is generated which is then passed through a second linear polarizer (identical to the first) rotated to be cross-polarized from the initial polarization. The projection of the lasing signal through the second polarizer was then passed into the same spectrometer and camera as used in the static experiment. A homebuilt LabView program was used for data acquisition.

**Transient reflectance**. Transient Reflectance measurements were done on a homebuilt transient reflectance setup, pumped by Ti:Sapphire amplifier (KM Labs, Wyvern 1000-50 operating at 10 kHz). A high-speed linear array detector (AVIIVA EM4, EV71YEM4CL1014-BA9, e2v) was used in conjunction with LabView for data acquisition.

**Atomic force microscopy**. A Bruker Dimension FastScan AFM in ambient conditions was used for all atomic force microscopy measurements. In COMSOL FEM simulations, we searched for modes at 2.408 eV (525 nm) using experimental geometry from AFM. As comparisons, we carried out simulations for equilateral triangle cross-sections with large lateral dimensions (see Supplementary Figure 8 and Supplementary Figure 9).

## Data availability

All relevant data are available from the authors upon request.

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

## Acknowledgements

X.-Y.Z. acknowledges the US Department of Energy, Office of Energy Science Grant DE-SC0010692 for supporting the time-resolved lasing and the transient reflectance experiments, the Vannevar Bush Faculty Fellowship of the US Department of Defense Grant ONR N00014-18-1-2080 for supporting the time-integrated photoluminescence experiments, and the Energy Frontier Research Center of the US Department of Energy Grant DE-SC0019443 for supporting the EM modeling. A.P. acknowledges support from the National Natural Science Foundation of China (Grant numbers 51525202 and 61574054) for supporting the growth of NW samples. The authors thank Prakriti Joshi and Skyler Jones for growing the CsPbBr3 crystals used in the transient reflectance measurements and Kihong Lee and Xinjue Zhong for help with atomic force microscope imaging.

## Author contributions

A.P.S., M.S.S. and X.-Y.Z. conceived and initiated this work. A.P.S. and M.S.S. performed time-resolved PL measurements. K.M. constructed the transient reflectance measurements with experimental help from F.L. X.W. synthesized the nanowires and A.P. supervised the synthesis. M.S.S. performed electromagnetic wave modeling with help from I.D. and M.L. A.P.S. and M.S.S. analyzed the data. A.P.S., M.S.S. and X.-Y.Z. wrote the manuscript. All authors read and commented on the manuscript.

## Additional information

**Competing interests:** The authors declare no competing interests.

