## [Peer Review File · Nature Communications]

Editorial Note: Parts of this peer review file have been redacted as indicated to remove third-party material where no permission to publish could be obtained.

Reviewers' comments:

Reviewer #1 (Remarks to the Author):

The manuscript by Schlaus et al. presented an investigation on the lasing mechanism in CsPbBr₃ nanowires using time resolved optical spectroscopy. The authors concluded that lasing originated from stimulated emission of electron-hole plasma rather than excitons. The manuscript elucidated important dynamic information on the lasing process, and therefore is of interest to the community. In addition, the manuscript is well written and the results are clearly illustrated. I support the publication of the manuscript in Nature communications.

I have a few minor suggestions:

1. Lasing modes that change as a function of carrier density and time (as depicted in Figure2) will present a real shortcoming in the CsPbBr₃ nanowire lasers. Could the authors elaborate on how such shortcoming could be overcome? Would it be possible to lower the threshold to below the Mott density?
2. Since lasing resulted from the coupling to a plasmon has also been observed in ZnO, why was the blue-shift of the laser modes not also observed in ZnO (discussion on page 5 and 6). Could the authors please clarify?
3. The caption of Figure 5 misses (b) and (c).

Reviewer #2 (Remarks to the Author):

The authors study photoluminescence and lasing in perovskite nanowires by ultrafast time-resolved photoluminescence spectroscopy and transient reflectance measurements. The claim that the mechanism behind the occurrence of lasing under pulsed excitation is stimulated emission from an electron-hole plasma, while earlier studies at low excitation densities and continuous excitation assigned the origin of lasing to the presence of excitons and exciton-polariton coupling. The development of lead halide perovskites and their application in optoelectronic applications is currently a very hot topic. Much of the underlying fundamental physics inherent to this class of semiconductors remains unknown up to now. In their study, the authors provide such fundamental insight into the origin of lasing under pulsed excitation of perovskite nanowires. The novelty of the work with respect to earlier reports lies in the claim that lasing occurs from a non-degenerate electron-hole plasma in the investigated material system. The paper is very physics-heavy and not easy to assess, however, in my opinion the results and data analysis support the conclusions of the authors and given the novelty of the work, the paper should be considered for publication. My main concern is that the paper is focused on just one material system, here the CsPbBr₃, and it is unclear to me why this system was chosen and not any other metal halide perovskite. This questions also the general importance and applicability of the results to other metal halide perovskite systems. The paper does not provide any comparison or insight that could support that the physics observed here are relevant also for other perovskites and not just in this particular case. I suggest the authors address this issue in a revision. Furthermore, the manuscript suffers in several instances from poor grammar and typos. Proofreading by a native speaker is required.

Reviewer #1

The manuscript by Schlaus et al. presented an investigation on the lasing mechanism in CsPbBr₃ nanowires using time resolved optical spectroscopy. The authors concluded that lasing originated from stimulated emission of electron-hole plasma rather than excitons. The manuscript elucidated important dynamic information on the lasing process, and therefore is of interest to the community. In addition, the manuscript is well written and the results are clearly illustrated. I support the publication of the manuscript in Nature communications.

Thanks.

I have a few minor suggestions:

1. Lasing modes that change as a function of carrier density and time (as depicted in Figure2) will present a real shortcoming in the CsPbBr₃ nanowire lasers. Could the authors elaborate on how such shortcoming could be overcome? Would it be possible to lower the threshold to below the Mott density?

The reviewer was right that this is indeed a challenge of the nanowire laser platform. While the Mott density is intrinsic to a particular material system and cannot be changed, the obstacles in the NW laser may be overcome by engineering approaches. We now add the follow to the last paragraph:

“While these findings reveal fundamental limitations of a NW as a stable lasing platform, there can be engineering approaches to overcome these obstacles. Examples include i) employing long excitation pulses and feedback in pumping power to stabilize the shifting lasing modes and ii) coupling the NW to an external optical cavity or phononic structure to select a narrow lasing wavelength window while suppressing all other wavelengths. From a mechanistic perspective, the shifting lasing modes in a NW cavity serves as ultra-sensitive probes of many-body dynamics.”

2. Since lasing resulted from the coupling to a plasmon has also been observed in ZnO, why was the blue-shift of the laser modes not also observed in ZnO (discussion on page 5 and 6). Could the authors please clarify?

Most previous time-resolved studies on ZnO NW lasing either did not use sufficient time resolution or carried out measurements at room temperature. We suspect that phonon replicas and much larger LO phonon energies in these measurements on ZnO NWs prohibited these authors from observing or analyzing time-dependent blue-shifts in the lasing profile. In one publication from Wille et al.

[Redacted]

[*Nanotechnology* 2016, 27(22), 225702, Fig. 3b, reproduced above], there was evidence for blue-shift on the 10s ps time scale, but the authors did not provide an analysis or interpretation. In the revised manuscript (bottom of page 5), we added the following sentence and the new reference 37 to point this out.

“In published spectra from one time-resolved study on ZnO NWs, there was evidence of blue-shift on the 10s ps time scale but the authors did not provide an analysis or interpretation³⁷.”

3. The caption of Figure 5 misses (b) and (c).

Fixed.

Reviewer #2

The authors study photoluminescence and lasing in perovskite nanowires by ultrafast time-resolved photoluminescence spectroscopy and transient reflectance measurements. The claim that the mechanism behind the occurrence of lasing under pulsed excitation is stimulated emission from an electron-hole plasma, while earlier studies at low excitation densities and continuous excitation assigned the origin of lasing to the presence of excitons and exciton-polariton coupling. The development of lead halide perovskites and their application in optoelectronic applications is currently a very hot topic. Much of the underlying fundamental physics inherent to this class of semiconductors remains unknown up to now. In their study, the authors provide such fundamental insight into the origin of lasing under pulsed excitation of perovskite nanowires. The novelty of the work with respect to earlier reports lies in the claim that lasing occurs from a non-degenerate electron-hole plasma in the investigated material system. The paper is very physics-heavy and not easy to assess, however, in my opinion the results and data analysis support the conclusions of the authors and given the novelty of the work, the paper should be considered for publication.

Thanks.

My main concern is that the paper is focused on just one material system, here the CsPbBr₃, and it is unclear to me why this system was chosen and not any other metal halide perovskite. This questions also the general importance and applicability of the results to other metal halide perovskite systems. The paper does not provide any comparison or insight that could support that the physics observed here are relevant also for other perovskites and not just in this particular case. I suggest the authors address this issue in a revision.

All of the perovskite nanowire systems showed similar absorption/PL spectra and lasing thresholds [refs. 1, 15-20]. We choose vapor grown CsPbBr₃ NWs because of its crystalline quality and thermal stability, permitting measurements in a broad window of excitation densities without thermal damage (see second paragraph on page 2). Compared to their hybrid organic-inorganic counter parts, CsPbBr₃ possesses higher exciton binding energy and smaller exciton radius than those in the hybrid counter parts [new refs. 41, 42] This means the Mott thresholds in MAPbBr₃ and FAPbBr₃ should be lower than that in CsPbBr₃ and

lasing in all three can be attributed to stimulated emission from degenerate e-h plasmas. We have added the following paragraph on page 9 to clarify this point.

“While the present study focuses on CsPbBr₃ NWs, the conclusions likely apply to other lead halide perovskite systems. Compared to their hybrid organic-inorganic counter parts, CsPbBr₃ possesses higher exciton binding energy and, thus, smaller exciton radius than those in the hybrid counter parts^{41,42}. This means the Mott thresholds in hybrid lead bromide perovskites are lower than that in CsPbBr₃ and lasing in these hybrid systems is also expected to be due to stimulated emission from a nondegenerate electron-hole plasma coupled with plasmons.”

Furthermore, the manuscript suffers in several instances from poor grammar and typos. Proofreading by a native speaker is required.

Fixed.

REVIEWERS' COMMENTS:

Reviewer #1 (Remarks to the Author):

The authors have address all my comments satisfactorily .

Reviewer #2 (Remarks to the Author):

I have re-assessed the manuscript by Schlaus et al. and evaluated the changes the authors have made in the revised version. I consider them sufficient and support publication of the manuscript. Few minor issues: the paragraph added in response to my comment, page 9 line 252 – 257, appears to be misplaced. I believe this paragraph should be moved to after the summary and conclusions. Furthermore, while the authors have managed to correct several typos and grammatical issues, some remained. Further correction on the editorial level may be required.